# Federated Progressive Sparsification
# (Purge-Merge-Tune)+

**Dimitris Stripelis**
Information Sciences Institute
University of Southern California
Marina Del Rey, CA 90292
stripeli@isi.edu

**Umang Gupta**
Information Sciences Institute
University of Southern California
Marina Del Rey, CA 90292
umanggup@isi.edu

**Greg Ver Steeg**
Information Sciences Institute
University of Southern California
Marina Del Rey, CA 90292
gregv@isi.edu

**José Luis Ambite**
Information Sciences Institute
University of Southern California
Marina Del Rey, CA 90292
ambite@isi.edu

## Abstract

We present *FedSparsify*, a sparsification strategy for federated training based on progressive weight magnitude pruning, which provides several benefits. First, since the size of the network becomes increasingly smaller, computation and communication costs during training are reduced. Second, the models are incrementally constrained to a smaller set of parameters, which facilitates alignment/merging of the local models, and results in improved learning performance at high sparsity. Third, the final sparsified model is significantly smaller, which improves inference efficiency. We analyze *FedSparsify's* convergence and empirically demonstrate that FedSparsify can learn a subnetwork smaller than a tenth of the size of the original model with the same or better accuracy compared to existing pruning and no-pruning baselines across several challenging federated learning environments. Our approach leads to an average 4-fold inference efficiency speedup and a 15-fold model size reduction over different domains and neural network architectures.

## 1 Introduction

Federated Learning [1, 2, 3, 4] has emerged as the standard distributed machine learning paradigm to train neural networks without sharing data. Each data source (client) trains the model on its private data and sends only its locally-trained model parameters (e.g., gradients, weights) to a central server. We are interested in reducing the communication cost during federated training and obtaining small models for fast inferences in resource-constrained devices [5].

Previous methods to speed up training and reduce model size include knowledge transfer [6], neural architecture search [7], and quantization [8, 9]. Inspired by model pruning in centralized training [10, 11], we propose *FedSparsify*, an iterative federated pruning procedure that progressively sparsifies model parameters during training. Our method simultaneously learns smaller neural networks for faster inference (and training) and reduces training communication costs by decreasing the total number of model parameters exchanged between the clients and the server.

We systematically compare *FedSparsify* to existing pruning techniques, including those that prune the model at the pre-training/initialization stage [12, 13], or dynamically through aggressive pruning

Workshop on Federated Learning: Recent Advances and New Challenges, in Conjunction with NeurIPS 2022 (FL-NeurIPS'22). This workshop does not have official proceedings and this paper is non-archival.

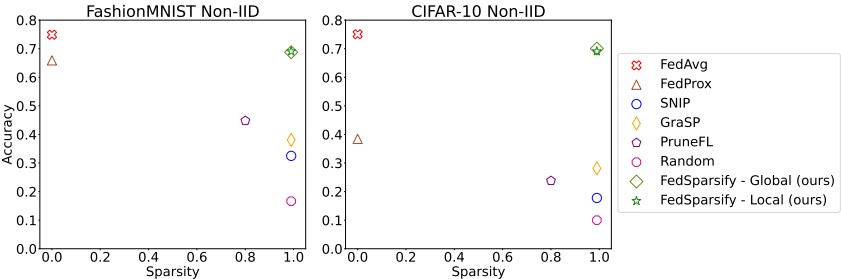

Figure 1: Test set accuracy for federated training with and without sparsification on the FashionMNIST and CIFAR-10 domains with Non-IID data distribution over a federation of 10 clients (99% sparsity).

and model regrowing during training [14], as well as no-pruning baselines [2, 15, 16]. *FedSparsify* learns sparsified models of similar performance to no-pruning methods and outperforms alternative pruning methods (see Figure 1), with a 4-fold reduction in communication costs and 4-fold increase in model throughput (see Section 5). Our main contributions can be summarized as follows:

1. Introducing iterative model pruning/sparsification in federated learning settings.
2. Reducing communication and inference costs by achieving extreme sparsification.
3. Analyzing local and global models pruning schedules.
4. A theoretical analysis of iterative pruning convergence in federated settings.

## 2   Related Work

Federated model pruning has been investigated in the context of enhancing privacy guarantees using gradient sparsification [17, 18] and mitigating model poisoning attacks by pruning the top-k model updates in conjunction with gradient clipping [19]. Other approaches have investigated gradient compression and quantization for communication cost reduction [20, 21, 8, 22]. A recent study [23] has also analyzed the convergence rate guarantees of pseudo-gradient sparsification on client and server in environments with full-client participation. Most of these works focus on faster convergence and reducing communication costs by pruning or quantizing gradients. In contrast, we aim to learn sparse models by model pruning for faster inference while reducing communication costs. Other works, PruneFL [14] and FedDST [24], investigate dynamic model pruning. PruneFL starts with a pruned model and readjusts the sparsification mask by allowing the model to grow periodically. FedDST trains with fixed sparsity budget throughout federated training by following a dynamic schedule that allows the client model to regrow periodically pruned parameters. Our *FedSparsify* strategy follows iterative cycles of pruning and fine-tuning, with a gradually increasing sparsity.

## 3   FedSparsify: Federated Purge-Merge-Tune

*FedSparsify* uses weight magnitude-based pruning at the clients and/or the federation controller (though we can also support other model pruning approaches [5, 25]). We describe the main design choices of *FedSparsify* below. A detailed algorithm appears in section B of the appendix.

**Weight Magnitude-based Pruning [26].** Neural network models often have millions of parameters, but not all parameters influence the outcome/predictions equally. A simple and surprisingly effective proxy to identify weights with small effect on the final outcome is based on the weights' magnitude [26, 11]. Weights with magnitudes lower than some threshold can be removed or set to zero without penalizing performance. The threshold is defined based on the number of parameters to be pruned (or prune percent, $s_t$). We prune parameters whose weight magnitude is in the bottom-$s_t$% in an unstructured way, considering the magnitude of each parameter separately. Approaches that prune groups of parameters (i.e., structured pruning) based on magnitude are also possible (e.g. [25]).

**Pruning Schedule.** A critical step in our approach is how often and how many parameters to prune during federated training. Pruning too many parameters early in training can lead to irrecoverable damage to the performance [11], and pruning too late leads to increased communication costs. To balance too early and too late pruning, we prune iteratively, by gradually reducing the number of trainable parameters. Finetuning after pruning often improves the performance, and allows pruning of

more parameters while preserving performance [27, 11]. Therefore, model pruning at the end of each federation round is a natural choice since clients can finetune the aggregated pruned global model during the next federation round. Two pruning approaches are applicable, prune locally at the clients before aggregation (FedSparsify-Local), or globally at the server after aggregation (FedSparsify-Global). We explore both strategies. Once a parameter is pruned, it never rejoins training (i.e., no network/weight regrowth). Motivated by [27], we apply the following pruning schedule:

$$s_t = S_T + (S_0 - S_T) \left( 1 - \frac{F \lfloor t/F \rfloor - t_0}{T - t_0} \right)^n \tag{1}$$

where $t$ is the federation round, $s_t$ is the model's sparsification percentage, $S_T$ is the final sparsification, $S_0$ is the initial sparsification percentage, $t_0$ is the round at which sparsification starts, $T$ is the total number of rounds, and $F$ is the sparsification frequency (e.g., $F = 1$ sparsifies at every round, while $F = 5$ sparsifies every 5 rounds). The exponent $n$ controls the rate of sparsification, with a higher n leading to aggressive sparsification at the start of training, and a smaller n leading to more sparsification towards the end of the training. In the experiments we use $n = 3$.

**FedSparsify-Local.** Model pruning takes place at each client after local training is complete. Each client sends its model, $w_k$, to the server along with the associated sparsification binary masks, $m_k$. The server may aggregate the local models using FedAvg, However, as the number of clients increases, it is increasingly unlikely that a given weight will be zero for all clients. This results in slow sparsification rates. To address this, we aggregate local models based on our proposed Majority Voting scheme, where a global model parameter is zeroed out only if less than half of local models' masks preserve it. Otherwise, the standard weighted average aggregation rule applies. Formally:

$$[m]_i = \begin{cases} 1 & \text{if } \sum_k^N [m_k]_i \geq \frac{N}{2} \\ 0 & \text{otherwise} \end{cases} \qquad w = m \odot \left( \sum_k^N \frac{|\mathcal{D}_k|}{|\mathcal{D}|} w_k \right) \tag{2}$$

where $[\cdot]_i$ is the corresponding value of the parameter at the $i^{\text{th}}$ position. $w$ is the global model, $N$ is the number of clients participating in the current round, and $m_k$ is the local binary mask of client $k$.

**FedSparsify-Global.** Model pruning occurs at the server right after participating clients' local models have been merged and the new sparse structure is maintained throughout local training. Therefore, all clients update the same set of model parameters and weighted aggregation of local models using FedAvg or Majority Voting is identical, i.e., no disagreement in local/global mask.

FedSparsify-Global and FedSparsify-Local pruning differ on mask sharing. Global pruning shares the global mask with clients at each federation round, and clients do not update the mask. In contrast, in local pruning the clients prune the parameters and share theire local masks with the server, which are then aggregated using the Majority Voting rule.

## 4 Convergence Analysis

We show the theoretical convergence rate for *FedSparsify* when $|\mathcal{D}_k| = |\mathcal{D}|/N, \forall k$, i.e., equal weights for each client and participation ratio is 1. These relaxations are made to simplify the analysis but these are not critical to the proof. See [28, 18] regarding the treatment of partial participation at each round and [28, 14] for analysis with consideration of weighted average. Our result (Thm. 1) shows that the convergence rate for *FedSparsify* is $\mathcal{O}(\frac{1}{T})$, which is the same as that of FedAvg [28]. However, compared to the usual federated training with FedAvg, the bound for *FedSparsify* has an additional term, the magnitude of difference of weights before and after pruning. We provide proof of the theorem and discuss this additional term in Appendix D.

**Theorem 1.** *If assumptions D.1-D.7 hold and with learning rate, $\eta < (4\sqrt{2}LS^{3/2})^{-1}$, then the parameters obtained at the end of each federation round of FedSparsify algorithm satisfy*

$$\frac{1}{T} \sum_{t=1}^T \left\| m^{(t)} \odot \nabla f(w^{(t)}) \right\|^2 \leq +2\eta L \left( \left( 1 + 4L\eta S^2 \right) \sigma^2 + \left( 16L\eta S^3 \right) \epsilon^2 \right)$$

$$+ \frac{4}{T\eta S} \mathbb{E} \left[ f \left( w^{(1)} \right) - f \left( w^{(*)} \right) \right] + \frac{4}{T\eta S} \sum_{t=1}^T L_p \left\| w^{(t+1)} - w^{(t+1)} \odot m^t \right\|$$

where $w^{(t+1)} \odot m^{(t)} := \frac{1}{N} \sum_{k=1}^{N} w_k^{(t,S)}$, *i.e., parameters right before sparsification is done and* $w^{(*)}$ *is the optimal parameter of sparsity* $S_T$.

## 5 Evaluation

We compare *FedSparsify* against a suite of pruning algorithms that perform model sparsification at different stages of federated training, as well as no-pruning methods. (The code to reproduce the experiments is publicly available; we do not share it now for anonymity.)

**Baselines.** We compare our *FedSparsify-Global* and *FedSparsify-Local* approaches against pruning at initialization schemes that sparsify the global model prior to federated training (SNIP [12], GraSP [13]), and a dynamic pruning scheme that prunes during training and performs local model regrowth (PruneFL [14]). We also compare with a progressive sparsification scheme that iteratively prunes the global (server-side) model weights during training *at random*.

Pruning at initialization schemes (SNIP [12], GraSP [13]) construct a fixed sparse model prior to the beginning of federated training. Following previous work [14, 24], we apply the schemes in a federated setting by randomly picking a client at the start of training to create the initial sparsification mask, which is enforced globally throughout training.

For dynamic pruning, we compare against PruneFL [14], which tries to maximize the reduction of empirical risk per training time unit by identifying prunable and non-prunable weights based on the ratio of gradients magnitude over parameter execution time. We follow the training and pruning configurations suggested in the original work. At the start of training, we randomly pick a client from the federation and learn the initial pruning mask after completing 5 reconfigurations. We perform global mask readjustment every 50 rounds and set the sparsification ratio for mask readjustment at round $t$ to $s \times 0.5^{\frac{t}{1000}}$ with $s = 0.3$, which is the recommended value.

We also consider a random pruning baseline (Random) to demonstrate the importance of pruning only weights with the smallest magnitude. We apply a progressive sparsification scheme but instead of using weight magnitude as selection criteria, we remove parameters randomly.

For no-pruning baselines, we consider FedAvg with Vanilla SGD [2], FedAvg with Momentum SGD [15], referred to as FedAvg (MFL), and FedProx [16]. For FashionMNIST we compare against FedAvg with SGD and FedProx, and for CIFAR-10 against FedAvg (MFL) and FedProx. We evaluate the efficacy of all these schemes over several degrees of sparsification.

**Federated Models & Environments.** We use FashionMNIST and CIFAR-10 as benchmark datasets, with a 2-layer fully-connected network for FashionMNIST, and a 6-layer convolutional network for CIFAR-10 (118,282 and 1,609,930 trainable parameters, respectively). We create four federated environments for each domain based on data distribution (IID and Non-IID), and number of clients (10 and 100 clients). For Non-IID data distributions, we assign examples from only a subset of classes to each client [29]: 2 classes (out of 10) per client for FashionMNIST Non-IID, and 5 classes (out of 10) per client for CIFAR-10 Non-IID. In environments with 10 clients, all clients participate at every round. In environments with 100 clients, 10 clients are randomly selected at each round (0.1 participation rate). (See Appendix Section C for hyperparameters details).

**Evaluation Criteria.** We evaluate the trade-off between model sparsity and learning performance (i.e., accuracy) for the different model pruning strategies. Our primary goal is to develop federated training strategies that learn a global model with the highest achievable accuracy at high sparsification rates. We measure learning performance at different degrees of sparsity (Figures 2a, 2b) and model convergence with respect to federation rounds and global model size reduction (Figures 2c, 2d). We do not measure convergence in terms of computation/wall-clock time speed-up, since we do not employ any dedicated hardware accelerators to leverage sparse operations.

**FashionMNIST Results.** Figure 2a shows the performance of different methods at different sparsification rates in the FashionMNIST domain for 10 clients, on both IID and Non-IID data distributions. The more complex the learning environment is (cf. IID vs Non-IID), the lower the final accuracy of the global model is for both pruning and no-pruning schemes. All sparsification methods have similar performance at moderate sparsification (i.e., 0.8, 0.85) with 10 clients and IID distribution. However, as sparsification becomes more extreme (i.e., 0.95, 0.99) and the data distribution becomes more challenging (Non-IID), existing sparsification methods underperform and, in some cases, cannot

learn a global model of reasonable performance (e.g., SNIP and GraSP in Non-IID). Although SNIP and GraSP (sparsification ratio: 0.8) can learn a sparsified model by restricting the model training to a predefined sparsified network, they suffer a substantial performance drop when compared to our FedSparsify schemes. We attribute this performance degradation to the binary mask learned over the local dataset of a randomly selected client, which may not necessarily follow the global data distribution and hence lead to a large performance gap between IID and Non-IID environments. FedSparsify outperforms alternative pruning methods at high-levels of sparsity.

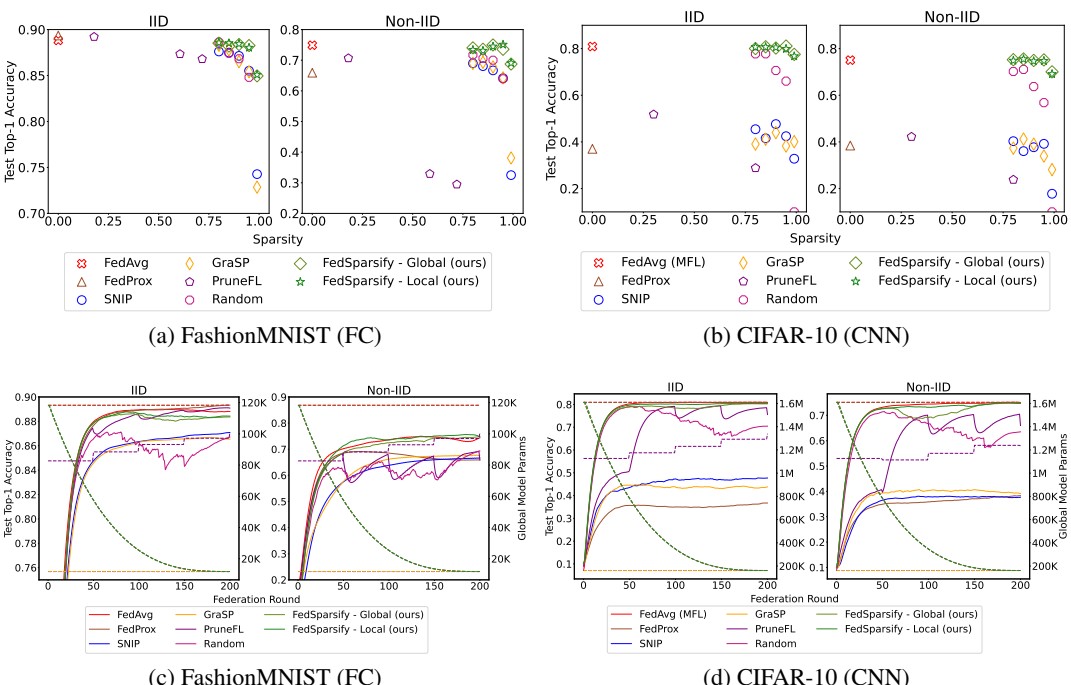

Figure 2: Sparsity vs. Accuracy (top row) and Federation Rounds vs. Accuracy (left y-axis) and Global Model Parameters Progression (right y-axis) (bottom row) for 10 clients.

Figure 2c shows the test accuracy (left y-axis, solid lines) and global model size in terms of total number of model parameters (right y-axis, dashed lines) as different approaches train (x-axis: federation rounds). For all sparsification schemes, the sparsity is set to 0.9 except for PruneFL, which is set to 0.3. FedAvg and FedProx have no sparsification, i.e., these are fully-parameterized models, and hence the global model has a constant size during training (top dashed lines). Similarly, pruning at initialization schemes are trained based on an already sparsified initial model and hence the global model size remains constant throughout training (bottom dashed lines). All progressive sparsification schemes (FedSparsify, Random) have a logarithmically decreasing global model size (mid-low decreasing dashed lines), while dynamic pruning (PruneFL) has a step-like increasing model size that is close to no-pruning methods. Our FedSparsify strategies have faster learning convergence in terms of federation rounds with performance comparable to or better than the fully-parameterized models. PruneFL's performance drops every 50 federation rounds due to the expansion of the model, with a stronger effect in the Non-IID environment. Similar to SNIP and GraSP, we attribute the degraded learning performance of PruneFL to the random client selection at the start of federated training to construct the initial sparsification mask. Finally, even though the Random scheme fails to preserve its learning performance towards the end of federated training, it is an effective pruning technique at the early stages of federated training when the sparsities are relatively small. We observed similar performances for all schemes in the more challenging federated environments of 100 clients for both IID and Non-IID distributions (see Figures 5a, 5c in Appendix E).

**CIFAR-10 Results.** We evaluate a 6-layer CNN architecture for CIFAR-10 across all four federated environments and five different sparsity levels (0.8, 0.85. 0.9, 0.95, and 0.99). As shown in Figures 2b and 2d FedSparsify outperforms by a large margin existing pruning at initialization and dynamic pruning approaches, while being able to learn sparse models at extreme sparsification rates (e.g., 0.9

- 0.99) with a learning performance similar and in some cases better than the no-pruning FedAvg (MFL) baseline (see Non-IID environment in Figure 2b). For all sparsifciation schemes in Figure 2b, convergence is shown at 0.9 sparsification, except for PruneFL that had 0.3 sparsifcation. We attribute the performance drop of pruning at initialization schemes to their need to remove a large proportion of the network's trainable weights at the beginning of training, a phenomenon which when also combined with the randomly assigned initial learning mask, leads to a degraded learning performance. Similarly, in the case of the dynamic PruneFL scheme that also relies on an initial randomly selected sparsification mask, even though it performs model regrowth during federated training, it is still not able to learn a sparse model of comparable performance. Interestingly, the Random pruning scheme is a strong baseline with comparable and often better performance compared to existing pruning methods. However, at extreme levels of sparsity, random pruning is not capable to learn, since at these levels of sparsification the remaining model weights are crucial and any random pruning may have an irreversible, negative result on the final model performance. The results on 100 clients (10% participation rate) are similar. FedSparsify outperforms alternative pruning methods with performance comparable to no-pruning methods. In the more challenging environment of CIFAR-10 Non-IID, FedSparsify-Global performs slightly better than FedSparsify-Local (see Figure 5 in the Appendix).

The goal of our sparsification strategy is to improve federated models' inference efficiency while at the same time being equally performant as no-pruning methods. Table 1 shows a comprehensive comparison of the performance of no-pruning (FedAvg) and sparsified models learned using our FedSparsify-Global approach in the non-IID environments with 10 clients for CIFAR-10 . Following previous work on benchmarking the inference efficiency of sparsified models [30, 31], we record the total number of batches (iterations) completed by the model within an allocated execution time and compute the number of processing items per second (throughput - items/sec), and the processing time per batch (ms/batch). Specifically, for the final model learned through the no-pruning (FedAvg) and FedSparsify-Global schemes we stress test its inference time by allocating a total execution time of 60 seconds with a warmup period of 10 seconds. Table 1 shows that the learned sparse CNN models can greatly improve inference efficiency when compared to fully parameterized networks. In particular, sparse models at 0.99 sparsity can provide a 4-fold improvement in terms of number of completed batches/iterations, latency and throughput, with only a small penalty ($\sim 7\%$) in model accuracy, while having a striking 56-fold model size compression and 2-4 reduction in communication costs (total number of parameters exchanged). The results on FashionMNIST over a fully connected network are similar (see Table 2 in the Appendix).

| Sparsity | Accuracy | Params | Model Size (MBs) | C.C. (MM) | Inf.Latency | Inf.Iterations | Inf.Throughput |
|---|---|---|---|---|---|---|---|
| 0.0 | 0.75 | 1,609,930 | 5.903 | 6,441 | 115 | 4,145 | 8,831 |
| 0.8 | 0.752 | 322,370 | 1.54 (x3.83) | 2,596 (x2.48) | 61 (x1.88) | 7,812 (x1.88) | 16,651 (x1.88) |
| 0.85 | 0.755 | 241,874 | 1.178 (x5.01) | 2,356 (x2.73) | 51 (x2.22) | 9,222 (x2.22) | 19,660 (x2.22) |
| 0.90 | 0.749 | 161,377 | 0.802 (x7.35) | 2,116 (x3.04) | 43 (x2.65) | 10,975 (x2.64) | 23,399 (x2.64) |
| 0.95 | 0.751 | 80,881 | 0.415 (x14.19) | 1,875 (x3.43) | 32 (x3.54) | 14,682 (x3.54) | 31,306 (x3.54) |
| 0.99 | 0.7 | 16,484 | 0.104 (x56.61) | 1,683 (x3.82) | 27 (x4.28) | 17,707 (x4.27) | 37,763 (x4.27) |

Table 1: Comparison of sparse (*FedSparsify-Global*), and non-sparse (*FedAvg*) federated models in the CIFAR-10 Non-IID environment with 10 clients. Values are measured based on the model learned at the end of federated training for 200 federation rounds. Sparsified models are learned using FedSparsify-Global. Sparsity 0.0 represents FedAvg. C.C.: communication cost in millions (MM) of parameters exchanged. Inference efficiency is measured by the mean processing time per batch (Inf.Latency - ms/batch), the number of iterations (Inf.Iterations), and processed examples per second (Inf.Throughput - examples/sec). Values in parenthesis show the reduction factor (model size, communication cost and inference latency) and increase/speedup factor (inference iterations and throughput) compared to no-pruning.

## 6   Conclusion

Scaling federated training is still a challenge, and it becomes more critical when training increasingly bigger models. In this work, we introduced FedSparsify, a novel pruning approach for federated training that progressively sparsifies a fully parameterized network, at the server *FedSparsify-Global* or at the clients *FedSparsify-Local*. Our iterative process of pruning and tuning produces highly

sparse subnetworks with learning performance similar to their non-sparse counterparts. At the same time, our process leads to a 4-fold improvement in model's inference efficiency, 4-fold reduction in the overall federated communication cost and a 15-fold model memory footprint reduction. In future work, we will explore performance improvements by using structured pruning approaches or by applying layer-specific thresholds. Recent works have also shown improved client-level privacy guarantees during federated training through gradient pruning [17]. We also plan to analyze the privacy gains of our federated sparsification approach and investigate whether we can improve privacy guarantees through stochastic model pruning approaches.

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

## A  Federated Optimization

In federated learning settings, the optimization goal is to find the set of optimal model parameters that minimizes the global objective function:

$$w^* = \operatorname*{argmin}_w \sum_{k=1}^{N} \frac{|\mathcal{D}_k|}{|\mathcal{D}|} f_k(w, \mathcal{D}_k) \quad (|\mathcal{D}| = \sum_k |\mathcal{D}_k|) \tag{3}$$

Where $f_k$ is the (local) objective function evaluated on each client's training dataset $\mathcal{D}_k$ and there are $N$ clients. Federated learning was introduced to train a neural network without aggregating private local data at a single location. In this work, we consider a federated learning environment [2] consisting of a central server and $N$ participating clients. The clients collaboratively train a machine learning model on their local private training datasets, $\mathcal{D}_k$. This federated environment is commonly referred to as *star-topology*. In this paper, we focus on this common centralized federated learning approach, even though other topologies exist as well [32, 33]. The server orchestrates the execution of the federation. Each client receives the global model from the server, $w$, and trains the model for an assigned number of local iterations. Depending on the number of participating clients and their availability, the server may delegate learning tasks to all the clients or a subset. The ratio of clients selected for the task out of the total number of clients is called *participation ratio* [34]. Upon completion of the assigned tasks, the server aggregates clients' local model parameters and computes a new global model.

One of the most popular approaches to aggregate clients' models is through a weighted model average with weights set proportional to the training samples used by the respective client. This approach is known as *FedAvg* [2] and clients train their local model using stochastic gradient descent (SGD). However, subsequent works have applied different update strategies during local training, such as MomentumSGD [3], also referred to as MFL, and FedProx [16], which introduces a proximal term to penalize the deviation of the local model from the global model. The works of [35, 36] interpret the local model updates generated by the clients as "pseudo-gradients" and propose aggregation strategies that are very similar to adaptive optimization techniques.

## B  FedSparsify Algorithm

We present the execution of FedSparsify-Global and FedSparsify-Local federated pruning methods in Algorithm 1. The server, i.e., `Server` procedure, is responsible to orchestrate the execution of the federation and delegate the training/learning tasks to the clients. Clients optimize locally on their

local dataset, i.e., `Client` procedure, the global model received by the server. Model purging/pruning is handled by the `purging_mask` function, either globally at the server (FedSparsify-Global) or locally at each client (FedSparsify-Local). When FedSparsify-Local is applied clients need also to share their local masks with the server and the server merges models through Majority Voting. When no sparsification is used FedSparsify-Global is equivalent to FedAvg.

---

**Algorithm 1:** `FedSparsify`. Global model $w$ and global mask $m$ are computed from $N$ participating clients, each indexed by $k$, at round $t$ out of a total number of $T$ rounds; $E$ is the local training epochs; $s_t$ is the sparsification percentage of model weights; $\mathcal{B}$ is the total number of batches per epoch; $\eta$ is the learning rate; $g_k^{(i)}$ denotes gradient of $k^{\text{th}}$ client's objective with parameters $w_k^{(i)}$. If no sparsification is used *FedSparsify-Global* is equivalent to FedAvg.

---

**Procedure** *Server($w^{(1)}, m^{(1)}$)*:

   **for** $t = 1$ **to** $T$ **do**

      **if** *FedSparsify-Global* **then**

         **for** $k = 1$ **to** $N$ **do**

            $w_k^{(t)} = Client(w^{(t)}, m^{(t)}, E, null)$

         $w^{(t+1)} = \sum_{k=1}^{N} \frac{|\mathcal{D}_k|}{|\mathcal{D}|} w_k^{(t)}$    $m^{(t+1)} = purging\_mask(w^{(t+1)}, s_t)$

         $w^{(t+1)} = w^{(t+1)} \odot m^{(t+1)}$

      **if** *FedSparsify-Local* **then**

         **for** $k = 1$ **to** $N$ **do**

            $w_k^{(t)}, m_k^{(t)} = Client(w^{(t)}, m^{(t)}, E, s_t)$

         $(w^{(t+1)}, m^{(t+1)}) :=$ merge params using Eq. 2

   **return** $w^{(t+1)}$

**Procedure** *Client($w, m, E, s_t$)*:

   $w_k^{(0)} = w$

   $S = E * \mathcal{B}$

   **for** $i = 0$ **to** $S$ **do**

      $w_k^{(i+1)} = w_k^{(i)} - \eta g_k^{(i)} \odot m$

   **if** *FedSparsify-Local* **then**

      $m_k = purging\_mask\left(w_k^{(S)}, s_t\right)$

      **return** $\left(w_k^{(S)}, m_k\right)$

   **return** $w_k^{(S)}$

---

# C  FedSparsify Tuning

In this section, we discuss the hyperparameters used to conduct the experiments in our study and the difference between Majority Voting and FedAvg as aggregation rules for FedSparsify-Local.

**Federated Hyperparameters.** Every federated model for both FashionMNIST and CIFAR-10 is trained for 200 rounds in total (cutoff-point) across all four federated environments. Each client trains for 4 local epochs with a batch size of 32. The learning rate is set to 0.02 for FashionMNIST and Vanilla SGD, and 0.005 for CIFAR-10 with the momentum attenuation factor set to 0.75. For FedProx, the proximal term $\mu$ is kept constant at 0.001. For all *FedSparsify-Local* and *FedSparsify-Global* experiments, sparsification starts at round 1 ($t_0 = 1$), initial degree of sparsification is 0 ($S_0 = 0$), sparsification frequency is 1 ($F = 1$, 1 round of tuning), and exponent is 3 ($n = 3$). During frequency value exploration, we observed that frequency values of $F = 1, 2$ behave similarly. However, for higher values of frequency (e.g., $F = 5, 10, 15, 20$), i.e., more rounds of fine-tuning, there is a big drop in the model performance when pruning takes place, since a larger number of weights is pruned in one shot. This phenomenon is also shown at Figure 3, where we explore different pruning frequencies. For FedSparsify-Local, we use Majority Voting as the aggregation rule of the local models, while for Random and FedSparsify-Global, we use FedAvg. The random seed for all the experiments is set to 1990. All experiments were run on a dedicated GPU server equipped with

4 Quadro RTX 6000/8000 graphics cards of 50 GB RAM each, 31 Intel(R) Xeon(R) Gold 5217 CPU @ 3.00GHz, and 251GB DDR4 RAM.

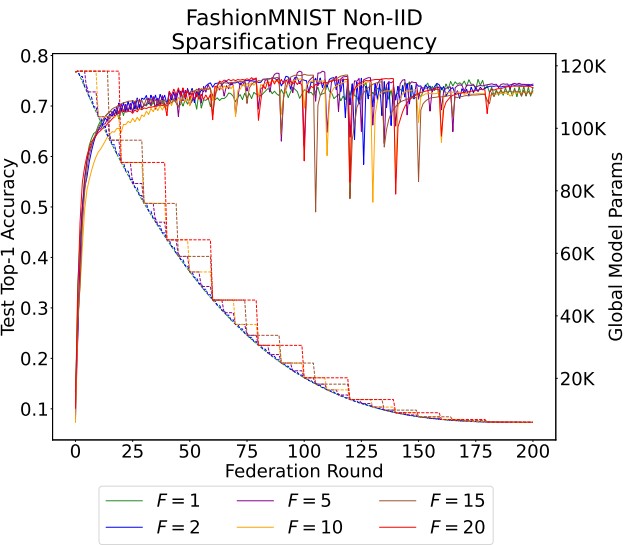

Figure 3: Sparsification frequency value exploration with FedSparsify-Global at 0.95 sparsity on FashionMNIST with 10 clients over Non-IID data distribution. Left y-axis and solid lines show accuracy, right y-axis show global model parameters progression. The higher the sparsification frequency, $F$, the bigger the drop in model performance.

**Majority Voting-based Aggregation**. In Figure 4 we show the learning performance (left y-axis) and global model parameters decrease (right y-axis) for the federated FashionMNIST model in a federated environment of 10 clients trained using the FedSparsify-Local sparsification schedule when Majority Voting and FedAvg are used as the aggregation rule of learners' local models. As it is shown (inset of the figure) at the beginning of training Majority Voting preserves the sparsity of the local models enforced by clients' local masks, while FedAvg resurrects some of these parameters.

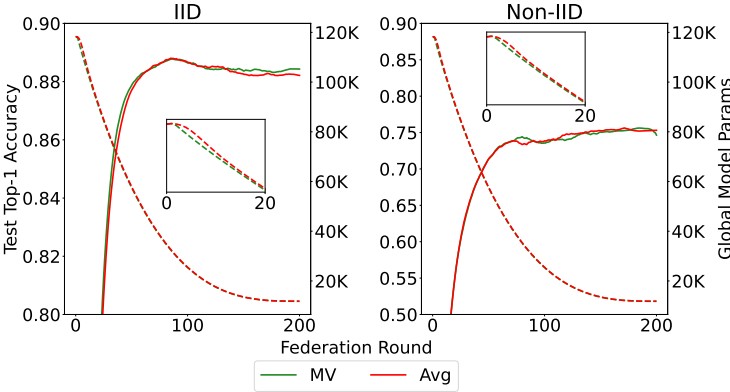

Figure 4: FedSparsify-Local with Majority-Voting (MV) as aggregation rule and FedSparsify-Local with Weighted Average (FedAvg/Avg) as aggregation rule on FashionMNIST with 10 clients over IID and Non-IID data distributions at 0.9 sparsity. Left y-axis and solid lines show accuracy, right y-axis and dashed lines show global model parameters reduction.

# D FedSparsify Convergence

## D.1 Further discussion of Thm. 1

Thm. 1 shows that the convergence bound for *FedSparsify* has an additional term compared to the usual federated training with FedAvg [28]. The difference is precisely the magnitude of weights that are pruned or removed. By noting that, $m^{(t)}$ describes the non-zero parameters in $t^{\text{th}}$ iteration and $w^{(t+1)} \odot m^{(t)} := \frac{1}{N} \sum_{k=1}^{N} w_k^{(t,S)}$, i.e., it is the aggregated parameters right before sparsification is done, we can further upper bound the difference by observing that

$$\left\| w^{(t+1)} - w^{(t+1)} \odot m^{(t)} \right\| \leq \left\| w^{(t+1)} \odot m^{(t)} \right\|$$

This is because $w^{(t+1)}$ is the sparsification outcome at the beginning of $t + 1^{\text{th}}$ iteration, and is obtained by zeroing out some parameters from $w^{(t+1)} \odot m^{(t)}$. The difference term in LHS is the magnitude of zeroed out parameters which is less than the magnitude of all parameters.

We can assume that the magnitude of neural network parameters is upper bounded by B (as assumed in [14]). However, this naive upper bound ignores that we purge parameters with the lowest magnitude in *FedSparsify-Global*. Therefore, we can compute a tighter bound for *FedSparsify-Global* by observing that $w^{(t+1)} \odot m^{(t)} - w^{(t+1)}$ will be 0 everywhere except for the indexes which are pruned to 0, i.e., the smallest entries, before $t + 1^{\text{th}}$ round. Note that exactly $\lfloor |w| \times s_{t+1} \rfloor - \lfloor |w| \times s_t \rfloor$ will be non zero, giving a tighter bound[1].

$$\left\| w^{(t+1)} - w^{(t+1)} \odot m^{(t)} \right\| \lesssim \left\| w^{(t+1)} \odot m^{(t)} \right\| (s_{t+1} - s_t) \lesssim \left\| w^{(t+1)} \right\| \frac{s_{t+1} - s_t}{1 - (s_{t+1} - s_t)}$$

In the case of FedSparsify-Local and majority voting, we remove parameters based on if most of the clients agree. Thus, the pruned parameter values are not necessarily the smallest, and the above discussed bound may not hold. In this work, we focused on removing a pre-defined percentage of parameters with the smallest magnitude. Based on the Thm. 1 more complicated strategies can be derived, such as removing parameters up to some threshold magnitude.

## Proof of Thm. 1

Proof Sketch of Thm. 1. To derive the proof, we make the same assumptions as earlier works of [14, 17]. Note that since we enforce sparsity or sparse structure found in previous iterations during client training and do not allow parameters to resurrect, we only need to show convergence of the average over $\left\| \nabla f(w^{(t)}) \odot m^{(t)} \right\|$ terms.

**Assumption D.1.** *Local objectives are smooth, i.e., $\|\nabla f_k(w_1) - \nabla f_k(w_2)\| \leq L\|w_1 - w_2\|$, $\forall w_1, w_2, k$ and some $L > 0$.*

**Assumption D.2.** *Global objective is lipschitz, i.e., $\|f(w_1) - f(w_2)\| \leq L_p \|w_1 - w_2\|$, $\forall w_1, w_2$ and some $L_p > 0$.*

**Assumption D.3.** *Client's stochastic gradients are unbiased, i.e., $\mathbb{E}[g_k(w)] = \nabla f_k(w)$, $\forall k, w$.*

**Assumption D.4.** *Local models have bounded gradient variance, i.e., $\mathbb{E}\|g_k(w) - \nabla f_k(w)\|^2 \leq \sigma^2$, $\forall k, w$.*

**Assumption D.5.** *The gradients from clients do not deviate much from the global model, i.e., $\|\nabla f(w) - \nabla f_k(w)\|^2 \leq \epsilon^2$, $\forall k, w$.*

**Assumption D.6.** *Time independent gradients, i.e., $\mathbb{E}\left[g_k^{(t_1)} g_k^{(t_2)}\right] = \mathbb{E}\left[g_k^{(t_1)}\right] \mathbb{E}\left[g_k^{(t_2)}\right]$, $\forall t_1 \neq t_2$.*

**Assumption D.7.** *Client independent gradients, i.e., $\mathbb{E}\left[g_{k_1}^{(t_1)} g_{k_2}^{(t_2)}\right] = \mathrm{E}\left[g_{k_1}^{(t_1)}\right] \mathbb{E}\left[g_{k_2}^{(t_2)}\right]$, $\forall k_1 \neq k_2$ and any $t_1, t_2$.*

*Proof.* The proof technique is similar to previous approaches that have demonstrated convergence for federated learning under different scenarios [18, 14, 28]. Proceeding similar to [18] and considering

---

[1]We use $\lesssim$ to indicate approximate inequality ignoring the issues that may occur due to floor operations.

$\mathbb{E}\left[f\left(w^{(t+1)}\right) \odot m^{(t)} - f\left(w^{(t)}\right)\right]$ we get —

$$\mathbb{E}[f(w^{t+1} \odot m^t) - f(w^t)] \leq \mathbb{E}\langle \nabla f(w^t), w^{t+1} \odot m^t - w^t \rangle$$
$$+ \frac{L}{2}\mathbb{E}\left\|w^{t+1} \odot m^t - w^t\right\|^2 \tag{4}$$

Considering the first term from above,

$$\mathbb{E}\langle \nabla f(w^t), w^{t+1} \odot m^t - w^t \rangle$$

$$= \eta\mathbb{E}\left\langle \nabla f(w^t), -\frac{1}{N}\sum_{k=1}^{N}\sum_{i=0}^{S-1} g_k^{t,i} \odot m^t \right\rangle$$

$$= \eta\mathbb{E}\left\langle \overline{\nabla f(w^t)} \odot m^t, -\frac{1}{N}\sum_{k=1}^{N}\sum_{i=0}^{S-1} \overline{\nabla f_k(w_k^{t,s})} \odot m^t \right\rangle$$

$$= -\eta\left\|\overline{\nabla f(w^t)} \odot m^t\right\|^2 - \eta\left\|\frac{1}{N}\sum_{k=1}^{N}\frac{1}{S}\sum_{i=0}^{S-1} \overline{\nabla f_k(w_k^{t,i})}\right\|^2$$

$$+ \eta\left\|\overline{\nabla f(w^t)} \odot m^t - \frac{1}{N}\sum_{k=1}^{N}\frac{1}{S}\sum_{i=0}^{S-1} m^t \odot \overline{\nabla f_k(w_k^{t,i})}\right\|^2$$

$$\leq -\eta\left\|m^t \odot \overline{\nabla f(w^t)}\right\|^2 - \frac{\eta}{NS}\sum_{k=1}^{N}\sum_{i=0}^{S-1}\left\|m^t \odot \overline{\nabla f_k(w^{t,i})}\right\|^2$$

$$+ \frac{\eta L^2}{NS}\sum_{k=1}^{N}\sum_{i=0}^{S-1}\left\|w^t - w_k^{t,i}\right\|^2 \tag{5}$$

For the second term in Eq. 4, we can establish by using assumptions 4-7 that,

$$\mathbb{E}\left\|w^{t+1} \odot m^t - w^t\right\|^2 = \mathbb{E}\left\|\frac{1}{N}\sum_{k=1}^{N}\sum_{i=0}^{S-1} m^t \odot \overline{g_k^{t,i}}\right\|^2$$

$$\leq S\sigma^2 + \frac{S}{N}\sum_{k=1}^{N}\sum_{i=0}^{S-1}\mathbb{E}\left\|m^t \odot \overline{\nabla f_k(w^{t,i})}\right\|^2 \tag{6}$$

By repeating analysis similar to lemma 10 from [18], we can obtain the below result.

$$\mathbb{E}\left\|w^{t,i} - w^t\right\|^2 \leq 16\eta^2 S^2\left\|m^t \odot \overline{\nabla f(w^t)}\right\|^2$$
$$+ 16\eta^2 S^2\epsilon^2 + 4\eta^2 S\sigma^2 \tag{7}$$

Using Eq. 5, 6, and 7 and subsitituting in Eq. 4, we get

$$\mathbb{E}\left[f\left(w^{t+1}\right) \odot m^t - f\left(w^t\right)\right] \leq$$
$$\left(-\frac{\eta S}{2} + 8L^2\eta^3 S^4\right)\left\|m^t \odot \nabla f(w^t)\right\|^2$$
$$+ \left(\frac{\eta^2 LS}{2} + 2L^2\eta^3 S^3\right)\sigma^2 + \left(8L^2\eta^3 S^4\right)\epsilon^2 \tag{8}$$

Above result establishes bound for the weight updates during federated training round. However, pruning can further change the models output, but we can control / bound its affect due to the lipschitz assumption. We can write:

$$E\left[f\left(w^{t+1}\right) - f\left(w^{t+1}\right) \odot m^t\right] \leq$$
$$L_p \left\|w^{t+1} - w^{t+1} \odot m^t\right\| \tag{9}$$

Adding the two, we get —

$$\mathbb{E}\left[f\left(w^{t+1}\right) - f\left(w^t\right)\right] \leq$$
$$\left(-\frac{\eta S}{2} + 8L^2\eta^3 S^4\right) \left\|m^t \odot \nabla f(w^t)\right\|^2$$
$$+ \left(\frac{\eta^2 LS}{2} + 2L^2\eta^3 S^3\right)\sigma^2$$
$$+ \left(8L^2\eta^3 S^4\right)\epsilon^2 + L_p \left\|w^{t+1} - w^{t+1} \odot m^t\right\|$$

Summing over all the time steps, and noting that

$$\mathbb{E}\left[f\left(w^{t+1}\right) - f\left(w^t\right)\right] \geq \mathbb{E}\left[f\left(w^*\right) - f\left(w^t\right)\right]$$

gives the desired result. $\square$

## E    FedSparsify Evaluation

We show the evaluation of FedSparsify to other pruning and no-pruning schemes in the federated environments with 100 clients in Figure 5. In Figure 6 we show federated models convergence in terms of cumulative transmission (communication) cost across all four federated environments, i.e., 10 and 100 clients at IID and Non-IID environments with a sparsification rate of 0.9 for all sparsification schemes except for PruneFL, which is shown at 0.3 (recommended sparsity). In Table 2 we show a holistic comparison of sparse and non-sparse federated models' throughput, inference, size and communication cost for the FashionMNIST sparse and non-sparse federated models in the environment of 10 learners. To measure all reported inference times we used the publicly available DeepSparse library, https://github.com/neuralmagic/deepsparse.

**Federated Environments with 100 Clients.** In these environments, at every federation round the server randomly selected 10 clients (participation ratio 0.1) to participate in the next training round. The execution results for these environments for FashionMNIST are shown in Figures 5a and 5c, and for CIFAR-10 in Figures 5b and 5d. In both domains, Figures 5a and 5b, FedSparsify is able to learn highly performant models at extreme sparsification rates (e.g., 0.95, 0.99 sparsity) that greatly outperform other sparsified models learned through other sparsification baseline schemes, cf. FedSparsify to GraSP and SNIP at 0.99 sparsity in the Non-IID learning environments. An interesting outcome of this evaluation is the performance of FedSparsify in the CIFAR-10 Non-IID environment. There, models learned using FedSparsify perform slightly better when compared to their non-sparse counterparts. We attribute this phenomenon to the regularization effect that sparsifcation may have on fully parameterized neural network models [5]. When comparing model convergence with respect to federation rounds and global model size reduction, as it expected, the no-pruning methods (FedAvg, FedProx) can learn models of improved learning performance at the expense though of fully parameterized final models. On the contrary though, through FedSparify we can learn sparsified federated models of similar or comparable performance with an extremely reduced number of model parameters. Moreover, when comparing other pruning schemes to FedSparsify we can see that the rest of the schemes plateau during federated training, whereas FedSparsify's learning curve is increasing, e.g., FashionMNIST IID and CIFAR-10 Non-IID learning environments.

**Transmission Cost.** We measure transmission cost in terms of Megabits (Mbit) exchanged for all federated training rounds. For both FashionMNIST and CIFAR-10 all federated models were trained for a total number of 200 rounds, hence the partially completed lines. We plot the total transmission cost of each scheme for the 200 rounds. The transmission cost at each round is computed as the total

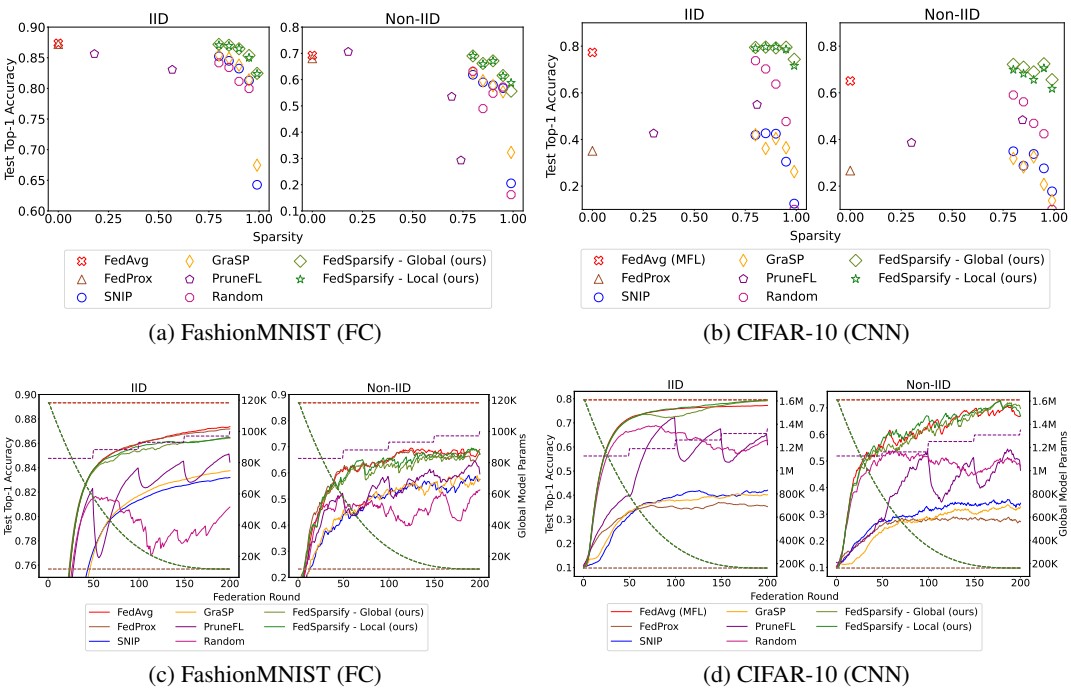

(a) FashionMNIST (FC)

(b) CIFAR-10 (CNN)

(c) FashionMNIST (FC)

(d) CIFAR-10 (CNN)

Figure 5: Evaluation for 100 clients with participation rate of 0.1 in terms of Sparsity vs. Accuracy (top row) and Federation Rounds vs. Accuracy (left y-axis) and Global Model Parameters Progression (right y-axis) (bottom row).

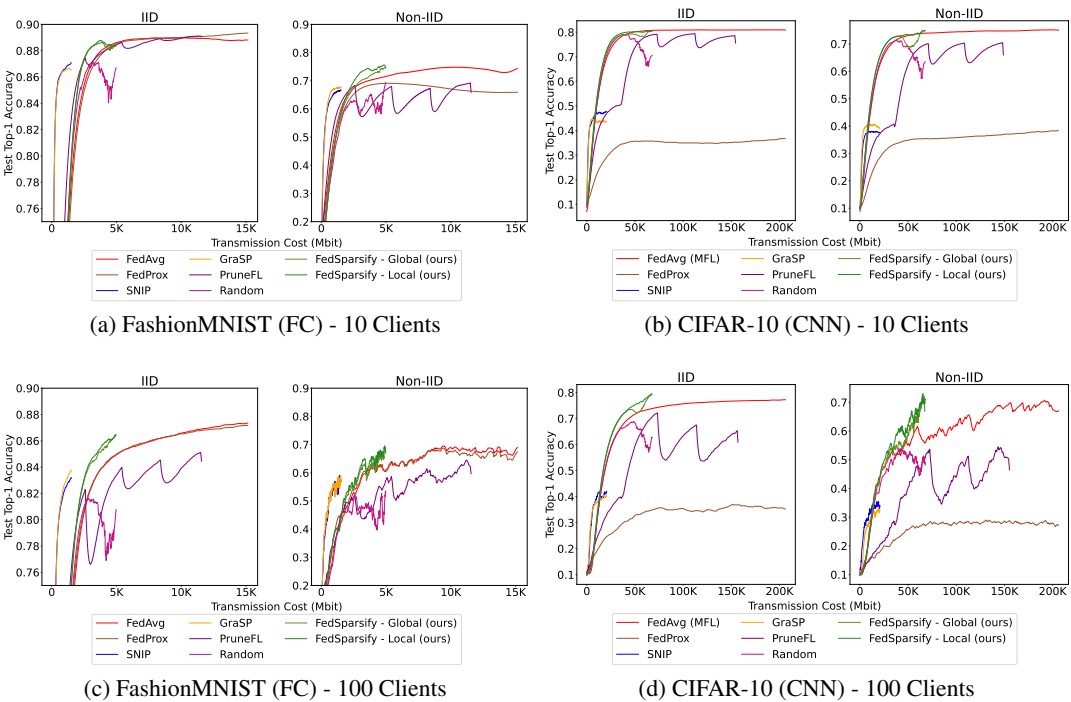

(a) FashionMNIST (FC) - 10 Clients

(b) CIFAR-10 (CNN) - 10 Clients

(c) FashionMNIST (FC) - 100 Clients

(d) CIFAR-10 (CNN) - 100 Clients

Figure 6: Transmission Cost vs. Accuracy for 10 clients (top row) and 100 clients (bottom row) over the federated training course of 200 federation rounds.

| Sparsity | Accuracy | Params | Model Size (MBs) | C.C. (MM) | Inf.Latency | Inf.Iterations | Inf.Throughput |
|----------|----------|--------|------------------|-----------|-------------|----------------|----------------|
| 0.0 | 0.7489 | 118,282 | 0.434 | 473 | 0.607 | 755,817 | 403,096 |
| 0.8 | 0.74 | 23,657 | 0.109 (x3.97) | 190 (x2.48) | 0.601 (x1.01) | 763,298 (x1.01) | 407,085 (x1.01) |
| 0.85 | 0.735 | 17,743 | 0.082 (x5.24) | 173 (x2.73) | 0.594 (x1.02) | 772,976 (x1.02) | 412,251 (x1.02) |
| 0.90 | 0.749 | 11,829 | 0.056 (x7.75) | 155 (x3.04) | 0.588 (x1.03) | 781,005 (x1.03) | 416,532 (x1.03) |
| 0.95 | 0.735 | 5,915 | 0.029 (x14.68) | 137 (x3.43) | 0.587 (x1.03) | 783,000 (x1.03) | 417,596 (x1.03) |
| 0.99 | 0.687 | 1,183 | 0.008 (x53.95) | 123 (x3.82) | 0.58 (x1.04) | 792,332 (x1.04) | 422,569 (x1.04) |

Table 2: Federated models comparison for FashionMNIST in the *Non-IID* environment of 10 clients. All recorded values are measurements from the model learned at the end of federated training for a total number of 200 federation rounds. All sparsified models represent the execution results of FedSparsify-Global and sparsity 0.0 of FedAvg. C.C. is an abbreviation for communication cost and captures the total number of exchanged parameters, expressed in millions (MM). Models inference efficiency is measured by mean processing time per batch (Inf.Latency - ms/batch), the number of iterations (Inf.Iterations), and processed items per second (Inf.Throughput - items/sec). Values in parenthesis show x-times reduction (for model size, communication cost and inference latency) and x-times increase/speedup (for inference iterations and throughput) compared to no-pruning.

number of clients participating at each round, multiplied by the total number of non-zero parameters received by the server at the beginning of the round (i.e., global model size), plus the total number of non-zero parameters uploaded to the server by all clients at the end of the round. We multiply this aggregated quantity by 32; we assume each parameter to be of 32-bits size. If the sparsification scheme exchanges binary masks with the server during federated training (e.g., FedSparsify-Local) then we also add to this quantity the total number of parameters of the original model, i.e., the size of the binary mask (1-bit parameters) is equal to the original model size without any sparsification. As it is also shown in Figures 6a and 6b for the FashionMNIST and CIFAR-10 domains, respectively, for the same number of Mbits exchanged between the clients and the server, FedSparsify is able to reach a higher learning performance when compared to other no-pruning (FedAvg) and pruning baselines (PruneFL). On the contrary though, pruned federated models learned through SNIP and GraSP schemes have a significantly reduced number of exchanged model parameters compared to the rest of the schemes, but they will require many more synchronization rounds to reach the performance of the other schemes. However, the same outcome does not hold for SNIP and GraSP in the CIFAR-10 domain where even with a small number of transmitted Mbits (e.g., 30k) they underperform all other pruning and no-pruning baselines. In both FashionMNIST and CIFAR-10 domains with 100 clients, Figures 6c and 6d, FedSparsify successfully learns a highly performant model that greatly outperforms all other approaches for the same number of exchanged Mbits (e.g., 5k Mbit in FashionMNIST, 50k Mbit in CIFAR-10).

