# OpenReview forum: "Federated Progressive Sparsification (Purge-Merge-Tune)+"
_NeurIPS.cc/2022/Workshop/Federated_Learning — FL-NeurIPS 2022 Poster_

### Official Review · Reviewer_YmVS · 2022-10-13
**A good experimental summary of sparsification methods**

This paper proposes simple sparsification methods for federated learning.
The proposed methods simply prunes the lower $s$-\% of the weights of deep neural networks (DNNs), either on the client-side or on the host-side.
The authors compared the proposed method with several existing sparsification methods, and show that the proposed method can sparsify DNNs at a cost of only a slight decay on test accuracy.

### Quality, Clarity
Overall, I found the paper is well-written and the main idea is easy to follow.

### Originality, Significance
Originality of the work is marginal, given that the proposed pruning strategy is a simple and a popular baseline in a non-distributed setting.
However, in my point of view, this fact will actually magnifies the significance of the paper.
That is, the current paper reports that such a simple baseline is quite effective also in distributed settings.
Thus, the current results will provide a rigid baseline for further researches on sparsification in distributed setting.
Providing a simple yet effective baseline will be an important contribution to the community.

---

### Official Review · Reviewer_3jAE · 2022-10-17
**Incremental work with good writing quality**

This paper presents FedSparsify, which conducts progressive pruning in a federated manner. FedSparsify leverages weight magnitude-based pruning, a carefully designed pruning schedule, and global majority vote-alike model aggregation. Both theoretical analysis and experimental results are given to demonstrate the effectiveness of FedSparsify.

Pros:
- The paper is well written.
- Both theoretical analysis and empirical results are given to demonstrate its effectiveness.

Cons:
- I'm not fully convinced by the motivation of this paper. Why do people want to conduct model sparsification in a collaborative manner (i.e., via federated learning)? Can't they simply train a smaller model?
- I'm also concerned by the novelty of this paper. It seems to be a simple extension of model sparsification to a federated scenario. The theoretical analysis seems also to be straightforward.
- The model size scales experimented on are also to be small. I'm not sure if such small models need to be compressed even in federated learning.

---

### Official Review · Reviewer_WJxS · 2022-10-17
**good paper**

This paper introduces two federated pruning scheme, FedSparsify-local and FedSparsify-global. Their proposed schemes are extensions of sparsification strategy at centralized setting, thus not too surprising from an algorithm perspective. The authors provided theorectical analysis and extensive experiments to study their proposed methods. I have the following questions:

1. In comparison to SNIP, do you apply SNIP in the FedSparsify-global framework, i.e., prune after aggregation? If so, it is worthy to understand why the advantage of SNIP in centralized setting does not translate to federated setting, or whether it is a reproduction issue.

2. Definition of the symbol ≲ is not provided in the theoretical analysis. D.1.

---

### Decision · Program_Chairs · 2022-10-20

Accept (Poster)